# Mediterranean Diet Adherence in Celiac Patients: A Nested Cross-Sectional Study

**DOI:** 10.3390/nu17050788

**Published:** 2025-02-25

**Authors:** Míra Zsófia Peresztegi, Zsolt Szakács, Zsófia Vereczkei, Eszter Dakó, Sarolta Dakó, Szilvia Lada, Klára Lemes, Miklós Holczer, Nelli Farkas, Judit Bajor

**Affiliations:** 1Medical School, University of Pécs, 7624 Pécs, Hungary; peresztegi.mira@pte.hu (M.Z.P.); miklosholczer@gmail.com (M.H.); 2First Department of Medicine, Medical School, University of Pécs, 7624 Pécs, Hungary; szakacs.zsolt@pte.hu; 3Institute for Translational Medicine, Medical School, University of Pécs, 7624 Pécs, Hungary; vereczkei47@gmail.com (Z.V.); nelli.farkas@aok.pte.hu (N.F.); 4Department of Sport Nutrition and Hydration, Institute of Nutritional Science and Dietetics, Faculty of Health Sciences, University of Pécs, 7621 Pécs, Hungary; 5Department of Surgery, Transplantation and Gastroenterology, Faculty of Medicine, Semmelweis University, 1082 Budapest, Hungary; dako.eszter@phd.semmelweis.hu (E.D.); dako.sarolta@gmail.com (S.D.); 6Directorate of Nursing Management and Professional Education, Albert Szent-Györgyi Health Center, University of Szeged, 6725 Szeged, Hungary; lada.szilvia@med.u-szeged.hu; 7Centre for Gastroenterology, Department of Medicine, Albert Szent-Györgyi Health Center, University of Szeged, 6725 Szeged, Hungary; lemes.klara@med.u-szeged.hu; 8Institute of Bioanalysis, Medical School, University of Pécs, 7624 Pécs, Hungary

**Keywords:** celiac disease, Mediterranean diet, gluten-free diet

## Abstract

**Background/Objectives**: The Mediterranean diet (MD) reduces cardiovascular risk, which is higher in celiac disease (CD). We aimed to investigate adherence to the MD in newly diagnosed CD patients, CD patients on a gluten-free diet (GFD), and in a non-celiac control group. Additionally, we aimed to establish an association between GFD and MD adherence. **Methods**: In this nested, cross-sectional Hungarian study, MD adherence was assessed using the Mediterranean Diet Score (MDS), and GFD adherence was assessed using the Standardized Dietitian Evaluation (SDE). **Results**: A total of 215 subjects were enrolled, 128 of which were CD patients on a GFD for a minimum of 1 year, 24 were newly diagnosed CD patients, and 63 were non-CD healthy control subjects. Although the control subjects had a higher mean MDS, the groups did not differ statistically significantly from each other (CD on GFD: 5.55 ± 1.57, newly diagnosed CD: 5.35 ± 1.81, controls: 6.05 ± 1.73; *p* > 0.05)—all groups had suboptimal scores. Both CD groups consumed fewer whole grains than the controls (*p <* 0.001). Adequate GFD adherence was associated with higher MDS (5.62 ± 1.54 vs. 4.71 ± 1.21, respectively; *p =* 0.009). **Conclusions**: Our study highlights the low adherence to MD in celiac patients with insufficient consumption of whole grains. Adherence to GFD is associated with better MD adherence, which underlines the role of dietary education during follow-up. Targeted nutritional counseling could improve the quality of diet in CD patients to reduce cardiovascular risk.

## 1. Introduction

Celiac disease (CD) is an immune-mediated disease that affects about 1% of the general population [1,2]. The only currently accepted therapy for CD is a lifelong gluten-free diet (GFD) [3,4,5]. While following a GFD, the autoimmune process attenuates, the mucosa regenerates, and absorption improves. Body weight increases but body composition does not always change favorably: fat-free mass tends to increase to a lesser extent, whereas fat content increases more significantly [6]. One of the contributing factors is the improperly implemented GFD that is high in calories, sugar, and fat, yet low in fiber. This can result in significant weight gain and unfavorable metabolic changes, such as dyslipidemia, hepatic steatosis, and insulin resistance [7,8,9,10]. Consequently, patients who are of normal weight or already overweight may face an increased cardiovascular risk. A global trend indicates that an increasing proportion of celiac patients are overweight or obese rather than underweight by the time of diagnosis [8,11]. This trend is partly explained by the growing prevalence of non-classical phenotypes of the disease and partly by the rising epidemic of obesity. For both known and unknown reasons, cardiovascular risk and the risk of certain metabolic disorders such as hepatic steatosis are elevated in CD [12,13,14,15,16,17,18]. Moreover, the unfavorable nutrient composition of gluten-free products, suboptimal dietary habits among patients, and often significant weight gain during the diet further aggravate the cardiovascular risk [19,20,21]. Although the GFD is essential for halting immunological processes and eliminating disease activity, in itself, it may not be sufficient to meet long-term health prevention goals [22,23].

From a cardiovascular and metabolic disease prevention standpoint, the Mediterranean diet (MD) is widely considered the most appropriate diet [24,25,26,27,28]. The concept of the MD was first defined by Ancel Keys. The Mediterranean dietary pyramid is based on dietary patterns observed in early 1960s Greece and southern Italy, where adult life expectancy was among the highest in the world and coronary heart disease, other diet-related chronic diseases, and obesity were among the lowest. The diet is characterized by a high intake of plant-based foods (vegetables, fruits, cereals, legumes, and nuts). Its favorable effects are attributed to antioxidant and anti-inflammatory properties found in foods rich in polyunsaturated fatty acids, polyphenols, and fiber while being low in ultra-processed products. After the Seven Countries Study, the MD became the ‘gold standard’ of diets for health maintenance and promotion [24]. Subsequent randomized trials provided high-level evidence that the MD offers advantages in both primary and secondary cardiovascular prevention [29,30]. Meta-analyses summarizing the results of these studies have also confirmed that stronger adherence to the MD correlates with lower all-cause mortality and reduced risk of cardiovascular diseases, including myocardial infarction, stroke, and cardiovascular death [27,31].

Multiple studies indicate that adherence to the MD in celiac patients is far from optimal [32,33,34]. Whole grains are an excellent source of dietary fiber, yet individuals with CD tend to consume insufficient fiber, which is a concern from both microbiome and cardiovascular risk perspectives. Two Spanish studies found that the diet of celiac patients is unbalanced; they consumed a diet higher in total fat, saturated fat, meat, added salt, and sugar and less fiber than non-celiac healthy individuals [35,36]. A case-control study in children also revealed higher fat intake and lower fiber intake among those with CD while MD adherence was suboptimal in both groups [37]. Another Italian study found poorer MD adherence among adults with CD and noted that they consumed more ultra-processed pasta and bread-based foods [38]. These findings highlight that strict GFD adherence alone does not guarantee an optimal diet for a healthy lifestyle.

Our study has two aims: to compare MD adherence (1) across CD patients on a GFD for a minimum of one year, newly diagnosed CD patients not following a GFD, and non-celiac subjects; (2) to compare between CD patients following adequate and inadequate GFDs.

## 2. Materials and Methods

### 2.1. Study Design

This is a nested cross-sectional study of a running prospective, multicenter study (ARCTIC, NCT05530070 [39]). This study was approved by the Scientific and Research Ethics Committee of the Hungarian Medical Research Council (27521-5/2022/EÜIG).

### 2.2. Study Population and Enrollment of Patients

This study included patients from three tertiary centers in Hungary: Semmelweis University, Budapest; the University of Pécs, Pécs; and the Albert Szentgyörgyi Medical University, Szeged. The patient inclusion was performed between November 2022 and April 2024. During participant recruitment, individuals aged 18 and older with CD, as well as non-celiac controls, were included in this study.

According to current guidelines, the diagnosis of CD in adults requires both positive serological and histological findings, which were verified for all included patients. For patients diagnosed in childhood, the ESPGHAN guidelines recommend a biopsy-free approach in cases where anti-tTG IgA levels are at least ten times the upper limit of normal and EMA-IgA positivity is confirmed in a second serum sample by appropriate testing. Newly diagnosed CD patients (before starting a GFD) and CD patients on GFD for a minimum of 1 year were included in this study. Participants who were assessed as not following the GFD using the Standardized Dietary Evaluation Method (SDE) were excluded from this study.

The control group consisted of non-celiac healthy individuals matched by age and gender. We recruited healthcare workers and medical students who met the inclusion criteria for the control group. Exclusion criteria for all participants included any acute illness, acute exacerbation of pre-existing chronic conditions, advanced chronic diseases (including heart failure, renal failure, and liver failure), systemic autoimmune diseases, malignancies, pregnancy, breastfeeding, or the presence of refractory CD.

Further methodological details of the ARCTIC trial were published elsewhere [39].

### 2.3. Dietary Assessment

GFD adherence was assessed by a trained dietitian using the Standardized Dietitian Evaluation (SDE) method [40]. ‘Excellent’ or ‘good’ assessments were considered to have an adequate GFD.

MD adherence was evaluated using the Mediterranean Diet Score (MDS), a validated 14-item questionnaire widely used as an indicator for healthy eating habits (Appendix A) [41]. It has a maximum score of 14 points and highlights consuming important components of the MD (vegetables, fish, and olive oil). A score of ≤8 was considered indicative of suboptimal adherence. We also analyzed MDS elements to determine whether participants’ intake was adequate or inadequate in each group: 1 point was regarded as adequate intake, and 0 and 0.5 points were considered inadequate.

### 2.4. Statistical Analysis

We summarized our data using descriptive statistics. For continuous variables, we reported either the mean and standard deviation (SD) or the median and interquartile range (IQR); for categorical variables, frequencies and percentages were calculated. To compare groups, we applied the Welch Two Sample *t*-test or the Kruskal–Wallis rank sum test for continuous variables as well as the Chi-squared test or Fisher’s exact test for categorical variables. When more than two groups were compared, the Bonferroni–Holm post hoc test was used to handle the false discovery rate. Differences were considered statistically significant if the *p*-value was less than 0.05. All analyses were performed in the R statistical environment (R version 4.4.2, R Core Team (2024), Vienna, Austria).

## 3. Results

This study included 215 subjects, of whom 128 were CD patients on a GFD for a minimum of 1 year, 24 were newly diagnosed CD patients, and 63 were non-CD healthy control subjects. The female predominance was approximately 80% across the groups (82%, 79%, and 84%, respectively). The baseline characteristics are presented in Table 1.

Among CD patients on a GFD for a minimum of 1 year, the average disease duration was 10 (±9) years, with the shortest duration being 1 year and the longest 47 years. A total of 48% of these patients qualified as ‘Excellent’ or ‘Good’ regarding GFD adherence.

### 3.1. Mediterranean Diet Adherence

Table 2 summarizes the results of MDS across the groups.

No significant difference was observed in the MDS across the groups. Among the 152 CD patients, only 11.2% (*n* = 17) achieved optimal adherence to the MD, whereas 19% (*n* = 12 out of 63) of the controls met this criterion. Both CD groups consumed significantly fewer whole grains than the control group (*p* < 0.001 for both comparisons). Newly diagnosed CD patients consumed legumes and nuts more often than the controls did (*p* = 0.017). They also reported consuming lean meats more frequently than participants in the other two groups (CD on GFD vs. new CD, *p* = 0.049; new CD vs. control, *p* = 0.043). Daily sugar intake was lower in the GFD group compared to newly diagnosed CD patients (*p* = 0.010). Concerning alcohol intake in line with MD guidelines (one glass a day for women and two for men, generally consumed with meals), the control group exhibited the most optimal pattern (GFD on CD vs. controls, *p* = 0.002; new CD vs. controls, *p* = 0.037).

No differences were found across the groups in the consumption of vegetables, fruits, fish, fatty meats, vegetable oils (e.g., olive or canola), high-fat non-fermented dairy products, or low-fat fermented dairy products (*p* > 0.05 for all comparisons).

The three groups (CD patients on a GFD, new CD patients, and non-celiac, healthy controls) were also compared based on whether their adherence to each element of the MD was adequate or inadequate (See Appendix A). Our findings were similar to the first comparison: both CD groups had a lower whole grain intake than the control group (CD on GFD vs. controls: *p* = 0.004; new CD vs. controls: *p* = 0.009). The new CD patients chose legumes more frequently in their diets than control individuals (*p* = 0.024). Sugar intake was more favorable in CD patients on a GFD than in new CD patients (*p* = 0.013). Wine consumption was more adherent to MD rules in the control group than in new CD patients (*p* = 0.050), and this difference was also observed in CD patients on a GFD; however, it did not reach the significance threshold (*p* = 0.054). We did not find any other notable differences in the other elements of the MD (*p* > 0.05 in all cases). The results are presented in Figure 1 and Appendix A.

### 3.2. Gluten-Free Diet Adherence and Mediterranean Diet Adherence

When comparing celiac patients with adequate versus inadequate GFD adherence, we found that those with adequate GFD adherence had significantly higher MD adherence (MDS 5.5 (4.5; 6.5) vs. 4.0 (4.0; 6.0); *p* = 0.009). Patients with inadequate GFD adherence consumed lean meats more frequently (*p* = 0.046); otherwise, no statistically significant differences were observed regarding the elements of the MD (Table 3).

## 4. Discussion

Our results show that the adherence to the MD of celiac patients does not significantly differ from that of the control group; however, their whole-grain fiber intake is notably lower. The overall MDS is suboptimal in all groups. Meanwhile, patients with adequate GFD adherence also exhibit better MD adherence.

Only a few studies have examined MD adherence among CD patients, with varying results. Interpreting these results is challenging, partly because the studies are heterogeneous and use different methods to assess MD adherence. For children, the KIDMED index is most commonly used [37,42], whereas adult studies employ various questionnaires (Mediterranean Diet Serving Score (MDSS), the Italian Mediterranean Index, etc.) [43,44,45]. Previous studies suggest that, overall, MD adherence in CD patients is far from optimal and is generally less favorable than in healthy individuals. One exception is a study of Moroccan adults in which celiac patients had better MD adherence than the healthy controls. Interestingly, eating disorders occurred less frequently in patients with good MD adherence [43]. In an Italian cross-sectional study involving 122 adults with CD and 100 healthy controls, MD adherence was assessed using the Italian Mediterranean Index. The Mediterranean Index was suboptimal in both groups but significantly worse among those with CD. The main reasons for poor adherence were the high consumption of potatoes and red and processed meats (foods not typically part of a MD) and lower fruit intake [45]. The largest pediatric study was conducted by Lionetti et al., who evaluated 120 children with CD and 100 controls. [37]. They found no significant difference in KIDMED scores (6.6 in CD vs. 6.8 in controls), and yet the diet of children with CD was less balanced; they consumed less fiber, more fat, more salty snacks, and fewer pseudocereals. Both groups had low vegetable, legume, and fish intake. Alarmingly, CD patients obtained 46% of their caloric needs from commercial gluten-free products. Similar results emerged in a study of Spanish children, which evaluated the nutritional adequacy of GFD: two-thirds had moderate or poor KIDMED scores, and their diets were unbalanced, containing more fat and less carbohydrates than recommended [42]. Another prospective cross-sectional study in children assessed the impact of MD adherence and physical activity on bone health. Higher adherence combined with regular physical activity was linked to better bone mineral density (Z-scores) and greater lean mass [46]. MD also has a beneficial effect on mental health. In Spyridaki’s study, although most participants had low adherence (MDSS < 14) (with a mean score of 9.44 ± 3.26 and 9.14 ± 3.07 for men and women, respectively, out of a total of 24 points), those with higher adherence had fewer psychopathological symptoms [44]. The beneficial effects of an MD have also been noted in other gastrointestinal (GI) conditions, including functional GI disorders and inflammatory bowel disease (IBD). IBD patients also appear to have poorer MD adherence compared to healthy controls, and there is a significant association between MD adherence and disease activity measured by fecal calprotectin, suggesting that the MD may help reduce intestinal inflammation [47].

In our study, we found suboptimal MD adherence among celiac patients, consistent with the literature. Although MDS values were lower in CD patients, the difference was not statistically significant (Table 2). The key difference between CD patients and healthy controls is the very low whole-grain fiber intake observed in both CD groups. Among those already following a GFD, this can be partly explained by the challenge of replacing forbidden grains (wheat, rye) with adequate high-fiber alternatives. Surprisingly, newly diagnosed CD patients—who are not yet on a long-term GFD—also consumed very few whole grains, which may be due to fiber intolerance caused by gastrointestinal complaints. Low fiber consumption is unfavorable for several reasons: it can negatively impact the microbiome and potentially increase the risk of cardiovascular and metabolic diseases. Prospective studies and clinical trials show that high dietary fiber and whole grain intake reduces all-cause and cardiovascular-related mortality, the incidence of type 2 diabetes and colorectal cancer, body weight, systolic blood pressure, and total cholesterol, with dose-response evidence supporting these findings [48].

An unexpected finding was the significant difference in legume and nut consumption between newly diagnosed CD patients and healthy individuals—a difference that was not observed in patients on a long-term GFD. The preference for lean meat among newly diagnosed CD patients may be an attempt to minimize digestive complaints rather than a purely health-conscious choice.

The dietary patterns of all three groups in our study—low consumption of fish, avocados, and olive oil and a preference for higher-fat dairy products—exhibit characteristics of a Western-type diet. Regarding moderate wine consumption, which is typical of the MD, the control group showed the most adherence to this pattern (consumption during meals, in recommended moderate amounts); their wine consumption was significantly higher than in both CD groups. Similar results were found in another study [45]. However, recently published studies recommend consuming no alcohol [49] because even a minimal amount was proven to be harmful. From that standpoint, the low wine consumption in the CD groups can be considered adequate.

When comparing newly diagnosed celiacs, CD patients on a GFD, and healthy controls based on adequate and inadequate MD adherence, the same results were obtained, except for the difference in lean meat consumption.

Our analysis of the relationship between GFD adherence and MD adherence was particularly instructive. Patients with inadequate GFD adherence displayed numerous other dietary imbalances (Table 3). For instance, none of the individuals with inadequate GFD adherence reported consuming avocados, fish, olive oil, canola oil, fruits, or whole grains, and their sugar intake was less favorable. The only exception was lean meat consumption, which was significantly higher in this group.

Despite the well-documented health benefits of the MD, Western dietary habits are becoming increasingly widespread worldwide. A large survey involving more than 10,000 Italians between 2019 and 2022 found that MD adherence is declining even in traditional Mediterranean regions [50]. Another Italian study revealed that CD patients with poorer MD adherence consume more ultra-processed foods [38], underlining the importance of proper follow-up care. It appears that inadequate diet adherence also correlates with insufficient ongoing clinical supervision [51]. Promoting the MD is crucial in all patient populations but should be especially emphasized in celiac patients. Weight gain often occurs on a GFD, and the unfavorable nutrient composition of many gluten-free products can lead to an imbalanced diet, further heightening the already increased cardiovascular risk in CD [16,20]. This risk needs targeted prevention during patient follow-up. One strategy is encouraging a healthy, nutritionally balanced GFD, ideally combining gluten-free requirements with MD principles [52]. Achieving this requires personalized dietary counseling and nutritional education programs [53].

### Strengths and Limitations

A major strength of our study is that, to our knowledge, it is the first to compare MD adherence in newly diagnosed celiacs, patients on a GFD, and healthy adults. Equally novel is our investigation of the relationship between MD adherence and GFD adherence.

The main limitations are the small sample size and the fact that our MD adherence assessment method cannot capture every detail of eating habits. Another limitation is the recollection bias, which is an inherent limitation of this study’s design. In addition, neither the MDS nor other commonly used MD adherence questionnaires are fully adapted to Hungarian dietary patterns—particularly regarding avocado consumption, which may lead to underestimation of MD adherence. Although avocados are now widely available in Hungarian supermarkets and popular among younger adults, regular consumption remains relatively low. Regarding statistical limitations, we did not adjust for multiplicity, as these results are intended for hypothesis-generating purposes.

## 5. Conclusions

The overall MDS remains lower in CD patients than in healthy individuals; however, MD adherence among celiac patients does not differ significantly from that of the control group. Our study highlights the dietary inadequacies of patients with CD—particularly their significantly lower whole grain intake compared to non-celiac controls. The consequent lower fiber consumption is unfavorable for several reasons: it can negatively impact the microbiome and potentially increase the risk of cardiovascular and metabolic diseases.

Our findings suggest that adherence to the MD in celiac patients requires improvement. Dietary counseling during the follow-up may improve not only the adherence to GFD but the adherence to MD as well. This dietary pattern has a role in the prevention of non-communicable diseases.

## Figures and Tables

**Figure 1 nutrients-17-00788-f001:**
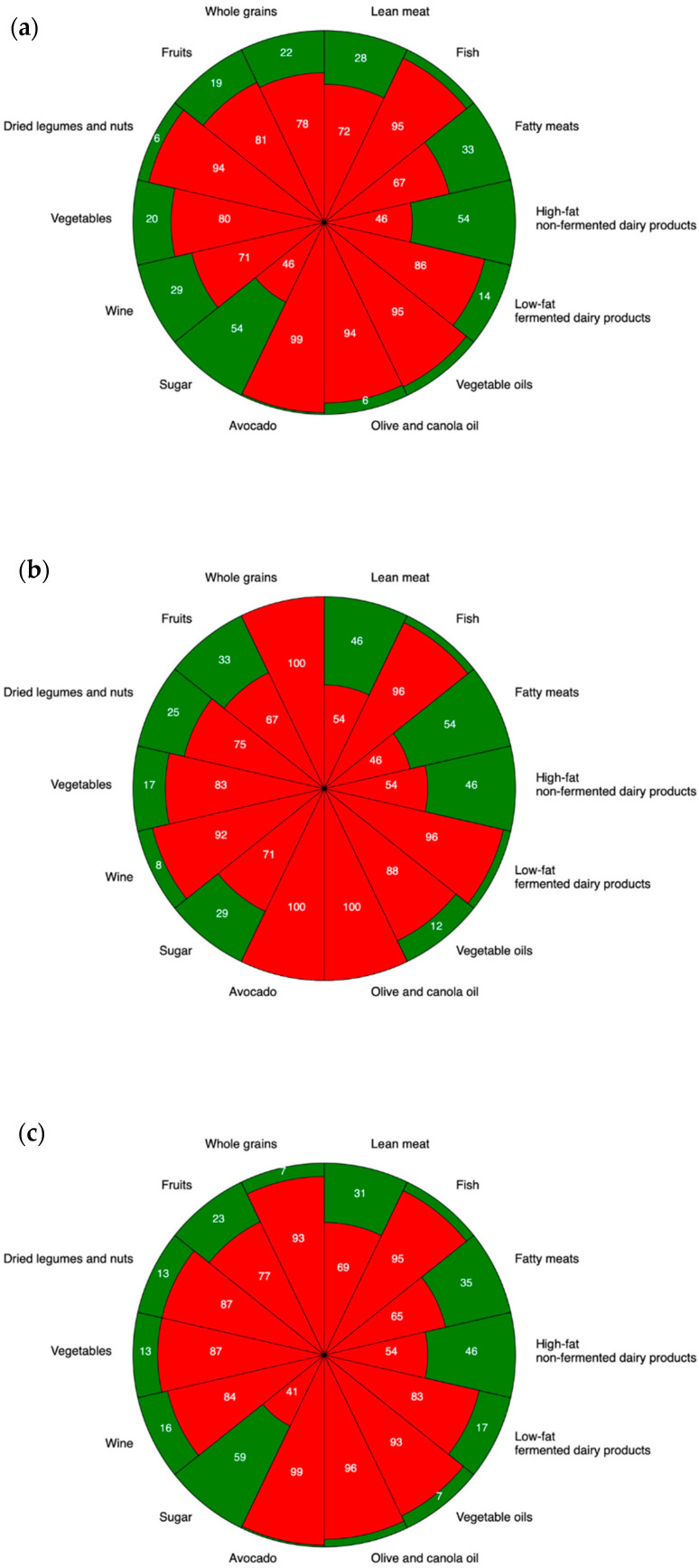
Adequate and inadequate Mediterranean diet adherence in controls (**a**), new CD patients (**b**), CD patients on a GFD (**c**). Inadequate adherence to MD, defined as an MDS of 0 or 0.5, is highlighted in red. The numbers are expressed as percentages. Adequate adherence to MD, defined as an MDS of 1, is highlighted in green.

**Table 1 nutrients-17-00788-t001:** Baseline characteristics of the study population. *p*-values indicate a difference across the three groups.

		CD Patients on GFD	Newly Diagnosed CD Patients	Control Subjects	*p*-Value
Number of subjects (*n*)		128	24	63	
Age at enrollment mean ± SD		36 (±13)	38 (±13)	32 (±11)	0.080
Sex *n* (%)	Women	105 (82%)	19 (79%)	53 (84%)	0.838
	Men	23 (18%)	5 (21%)	10 (16%)	
Diabetes *n* (%)		8 (6.2%)	1 (4.2%)	1 (1.6%)	0.384
BMI mean ± SD		23.8 (±5.3)	24 (±6.8)	23.6 (±4.1)	0.874

CD: celiac disease, BMI: Body Mass Index, SD: standard deviation. Statistical test: Kruskal–Wallis rank sum test, Chi-squared test, Fisher’s exact test as appropriate, level of significance: *p* < 0.05.

**Table 2 nutrients-17-00788-t002:** Mediterranean Diet Score across the groups.

		CD on GFD *n* = 128	New CD *n* = 24	Controls *n* = 63	CD on GFD vs. New CD	CD on GFD vs. Controls	New CD vs. Controls
					*p*-Value		
MDS Score median (IQR)		5.5 (4.5; 6.5)	5.0 (4.0; 6.1)	6.0 (5.0; 7.0)	0.300	0.081	0.070
Vegetables *n* (%)	0	17 (13%)	6 (25%)	8 (13%)	0.200	0.400	0.400
	0.5	94 (74%)	14 (58%)	42 (67%)			
	1	17 (13%)	4 (17%)	13 (20%)			
Legumes and nuts *n* (%)	0	56 (44%)	8 (33%)	39 (62%)	0.300	0.056	**0.017**
	0.5	55 (43%)	10 (42%)	20 (32%)			
	1	17 (13%)	6 (25%)	4 (6%)			
Fruits *n* (%)	0	24 (19%)	5 (21%)	13 (21%)	0.400	0.800	0.300
	0.5	74 (58%)	11 (46%)	38 (60%)			
	1	30 (23%)	8 (33%)	12 (19%)			
Whole grains *n* (%)	0	93 (73%)	16 (67%)	17 (27%)	0.200	**<0.001**	**<0.001**
	0.5	26 (20%)	8 (33%)	32 (51%)			
	1	9 (7%)	0 (0%)	14 (22%)			
Lean meat *n* (%)	0	17 (13%)	6 (25%)	8 (13%)	**0.049**	>0.999	**0.043**
	0.5	71 (56%)	7 (29%)	37 (59%)			
	1	40 (31%)	11 (46%)	18 (28%)			
Fish *n* (%)	0	95 (74%)	17 (71%)	42 (67%)	>0.999	0.500	>0.999
	0.5	27 (21%)	6 (25%)	18 (28%)			
	1	6 (5%)	1 (4%)	3 (5%)			
Fatty meat and processed meat *n* (%)	0	33 (26%)	2 (8%)	13 (21%)	0.091	0.600	0.150
	0.5	50 (39%)	9 (38%)	29 (46%)			
	1	45 (35%)	13 (54%)	21 (33%)			
High-fat, non-fermented dairy products *n* (%)	0	24 (19%)	5 (21%)	6 (9%)	>0.999	0.300	0.400
	0.5	45 (35%)	8 (33%)	23 (37%)			
	1	59 (46%)	11 (46%)	34 (54%)			
Low-fat fermented dairy products *n* (%)	0	49 (38%)	13 (54%)	27 (43%)	0.200	0.800	0.400
	0.5	58 (45%)	10 (42%)	27 (43%)			
	1	21 (17%)	1 (4%)	9 (14%)			
Vegetable oils *n* (%)	0	34 (27%)	9 (38%)	19 (30%)	0.200	0.800	0.300
	0.5	85 (66%)	12 (50%)	41 (65%)			
	1	9 (7%)	3 (12%)	3 (5%)			
Olive and canola oil *n* (%)	0	77 (60%)	17 (71%)	32 (51%)	0.600	0.400	0.200
	0.5	46 (36%)	7 (29%)	27 (43%)			
	1	5 (4%)	0 (0%)	4 (6%)			
Avocado *n* (%)	0	107 (84%)	18 (75%)	49 (78%)	0.500	0.600	0.800
	0.5	19 (15%)	6 (25%)	13 (21%)			
	1	2 (1%)	0 (0%)	1 (1%)			
Sugar *n* (%)	0	14 (11%)	7 (29%)	9 (14%)	**0.010**	0.700	0.086
	0.5	39 (30%)	10 (42%)	20 (32%)			
	1	75 (59%)	7 (29%)	34 (54%)			
Wine *n* (%)	0	87 (68%)	17 (71%)	26 (41%)	0.700	**0.002**	**0.037**
	0.5	21 (16%)	5 (21%)	19 (30%)			
	1	20 (16%)	2 (8%)	18 (29%)			

CD: celiac disease, MDS: Mediterranean Diet Score, IQR: interquartile range. All percentages were rounded to the nearest whole number. Statistical test: Kruskal–Wallis rank sum test, Chi-squared test, Fisher’s exact test as appropriate, level of significance: *p* < 0.05. Significant results are highlighted with bold numbers.

**Table 3 nutrients-17-00788-t003:** Mediterranean Diet Score of CD patients with adequate and inadequate GFD adherence.

		CD Patients with Adequate GFD Adherence *n* = 109	CD Patients with Inadequate GFD Adherence *n* = 17	*p*-Value
MDS Score median (IQR)		5.5 (4.5; 6.5)	4.0 (4.0; 6.0)	**0.009**
Vegetables *n* (%)	0	14 (13%)	3 (18%)	0.456
	0.5	83 (76%)	11 (64%)	
	1	12 (11%)	3 (18%)	
Legumes and nuts *n* (%)	0	44 (40%)	10 (59%)	0.344
	0.5	49 (45%)	6 (35%)	
	1	16 (15%)	1 (6%)	
Fruits *n* (%)	0	18 (16%)	6 (35%)	0.172
	0.5	64 (59%)	9 (53%)	
	1	27 (25%)	2 (12%)	
Whole grains *n* (%)	0	79 (73%)	14 (82%)	0.805
	0.5	23 (21%)	2 (12%)	
	1	7 (6%)	1 (6%)	
Lean meat *n* (%)	0	13 (12%)	3 (18%)	**0.046**
	0.5	65 (60%)	5 (29%)	
	1	31 (28%)	9 (53%)	
Fish *n* (%)	0	81 (74%)	12 (71%)	0.614
	0.5	22 (20%)	5 (29%)	
	1	6 (6%)	0 (0%)	
Fatty meat and processed meat *n* (%)	0	29 (27%)	4 (24%)	0.801
	0.5	44 (40%)	6 (35%)	
	1	36 (33%)	7 (41%)	
High-fat, non-fermented dairy products *n* (%)	0	17 (16%)	6 (35%)	0.131
	0.5	38 (35%)	6 (35%)	
	1	54 (49%)	5 (30%)	
Low-fat fermented dairy products *n* (%)	0	41 (38%)	8 (47%)	0.774
	0.5	51 (47%)	7 (41%)	
	1	17 (15%)	2 (12%)	
Vegetable oils *n* (%)	0	27 (25%)	7 (41%)	0.325
	0.5	75 (69%)	9 (53%)	
	1	7 (6%)	1 (6%)	
Olive and canola oil *n* (%)	0	64 (59%)	13 (76%)	0.390
	0.5	41 (37%)	4 (24%)	
	1	4 (4%)	0 (0%)	
Avocado *n* (%)	0	91 (83%)	15 (88%)	>0.999
	0.5	16 (15%)	2 (12%)	
	1	2 (2%)	0 (0%)	
Sugar *n* (%)	0	11 (10%)	3 (18%)	0.470
	0.5	33 (30%)	6 (35%)	
	1	65 (60%)	8 (47%)	
Wine *n* (%)	0	74 (68%)	13 (76%)	0.167
	0.5	17 (16%)	4 (24%)	
	1	18 (16%)	0 (0%)	

CD: celiac disease, GFD: gluten-free diet, MDS: Mediterranean Diet Score, IQR: interquartile range. All percentages were rounded to the nearest whole number. Statistical tests: Welch’s Two Sample *t*-test, Chi-squared test, Fisher’s exact test as appropriate; level of significance: *p* < 0.05. Significant results are highlighted with bold numbers.

## Data Availability

The data presented in this study are available within the article and the Appendix A. The raw data are available on request from the corresponding author.

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
