# Peer review of "Mediterranean Diet Adherence in Celiac Patients: A Nested Cross-Sectional Study"

_nutrients, 2025, doi:10.3390/nu17050788_

Round 1
Reviewer 1 Report
Comments and Suggestions for Authors
This study investigates the nutrition of patients with celiac disease and whether they are able to adhere to the Mediterranean diet which is recognized as one of the top three diets for reduction of cardiovascular risk which is of great importance to patients with elevated cv risk such as celiac patients. The manuscript is well written with adequate background presented in the introduction.
I have a few questions i believe the authors should address to further explain methodology
how exactly did u perform a nested control study with this many groups, and why?
Why would these patients adhere to the meditarrean diet to begin with, did they take some counceling about mediterranian diet prior to inclusion in the study or upon diagnosis?
In tables please write all p values with 3 decimal places
increase font in figure 1
In discussion it would be more appropriate to start with your results first.
Also you mention a lot western diet adherence, was this tested and how exactly?
The first sentence in the conclusion seems to state two opposite things, please rephrase.
You should discuss the low grain intake more in depth in the discussion section as it appears in the conclusion. I can suggest mentioning blue zones as well.
Is figure S1 really a figure?
Author Response
Comment 1 how exactly did u perform a nested control study with this many groups, and why?
Response 1
Thank you very much for taking the time to review this manuscript.
Currently, we have a running comprehensive project to study cardiovascular and metabolic risk in celiac disease (ARCTIC study).
https://pmc.ncbi.nlm.nih.gov/articles/PMC10503320/
https://pubmed.ncbi.nlm.nih.gov/39384235/
The project has three parts: 1) a case-control study, which compares newly diagnosed celiac patients and patients on a gluten-free diet to a non-celiac control group regarding body composition and cardiovascular risk-related metabolic parameters; 2) a prospective cohort study, which investigates how body composition and cardiovascular risk-related metabolic parameters change during a 1-year gluten-free diet started after the diagnosis of celiac disease; and 3) a randomized controlled study, which investigates how a structured, 1-year-long, repeated dietetic intervention influences body composition and cardiovascular risk-related metabolic parameters, compared to standard of care, in celiac disease patients following a strict gluten-free diet for at least 1 year. All these projects provide high-quality data on the participants' cardiovascular risk; however, the question of dietary adherence to the Mediterranean diet is out of the scope. This cross-sectional setting offers a cost-effective way to analyze the already-collected data from the three projects a new point-of-view to study dietary habits regarding the Mediterranean diet. We clarified this in the manuscript, as well.
Comment 2 Why would these patients adhere to the meditarrean diet to begin with, did they take some counceling about mediterranian diet prior to inclusion in the study or upon diagnosis?
Response 2
No dietary education related to the Mediterranean diet was provided in either group. When we talk about “adherence to the Mediterranean diet,” we are actually assessing the extent to which an individual’s diet aligns with the principles of the Mediterranean diet. This is the same approach present in all the publications cited on this topic.
An important message of our study is that although current guidelines for celiac patients emphasize that dietary consultations should focus on providing information about the gluten-free diet and ensuring adherence to it, compliance with the principles of the Mediterranean diet could offer additional health benefits to these patients.
The ongoing RCT arm of our ARCTIC study is investigating the effects of a dietary intervention focused on the Mediterranean diet. The results of this study will provide high-level evidence on this topic.
Comment 3 In tables please write all p values with 3 decimal places
Response 3
We thank you for your helpful suggestion. We have revised all p-values in the tables to be reported with three decimal places as requested.
Comment 4 increase font in figure 1
Response 4
The font size in Figure 1 has been increased to improve readability.
Comment 5 In discussion it would be more appropriate to start with your results first.
Response 5
We corrected the Discussion Section accordingly.
Comment 6 Also you mention a lot western diet adherence, was this tested and how exactly?
Response 6
We did not specifically assess adherence to the Western diet; however, the responses from the questionnaire measuring adherence to the Mediterranean diet indicate that the dietary patterns of the participants align with characteristics of a Western-style diet. The original sentence on this matter was indeed ambiguous, so we have revised it for clarity (Row 261-263): The dietary pattern of all three groups in our study — low consumption of fish, avocado, and olive oil, and a preference for higher-fat dairy products – exhibit characteristics of a Western type diet.
Comment 7 The first sentence in the conclusion seems to state two opposite things, please rephrase.
Response 7
We rephrased the first sentence accordingly: „The adherence to MD is low in CD patients, however, it does not differ significantly from that of the control group”.
Comment 8 You should discuss the low grain intake more in depth in the discussion section as it appears in the conclusion. I can suggest mentioning blue zones as well.
Response 8
Thank you for your suggestion. We have supplemented the text accordingly: “Low fiber consumption is unfavorable for several reasons: it can negatively impact the microbiome and potentially increase the risk of cardiovascular and metabolic diseases. Prospective studies and clinical trials show that high dietary fiber and whole grain intakes reduce all-cause and cardiovascular-related mortality, the incidence of type 2 diabetes and colorectal cancer, body weight, systolic blood pressure and total cholesterol, with dose-response evidence supporting these findings.” (Reynolds A, Lancet, 2019)
Comment 9 Is figure S1 really a figure?
Response 9 Thank you for your insight, S1 is actually a Table.
Again, thank you for all the comments and suggestions made. Please, let us know if any further changes or improvements are required.
Reviewer 2 Report
Comments and Suggestions for Authors
I think this is a nicely conducted study with a reasonable and straightforward message about the missed opportunity of following a beneficial diet in patients who are in need of those benefits.
I have some specific suggestions below, but an overarching comment is that I am unclear about the patient’s medical advice. The paper is written as though we are to assume all (CD) volunteers will have been advised to follow the MD as part of their treatment. The language throughout is generally presumptive of this and classifying their diets as being “inadequate” etc. implies they are failing at following a guideline. I am not sure this is the case though? If the patients have all been told they should follow the MD this should be stated explicitly. If they have not then I think it reframes the paper’s objectives and would encourage the authors to adjust some of the introduction / discussion to make this less like patients may not be following advice, and more about this being a possible missed opportunity that the medical community should consider adopting properly in treatment approach.
Other suggestions:
Abstract: Additional methodological information would be desirable, particularly where the participants in the study came from and also the sample sizes.
Methods (2.2): I appreciate the citation to more methodological information but please state in the current manuscript where the control group was recruited from. It would also be beneficial in the paper to state the headline exclusion criteria that apply to all participants (taken from the ARCTIC paper, e.g. heart failure, active cancers, refractory CD etc.).
I’d also like elaboration on the “newly diagnosed” group. At what point does study data collection occur with respect to their CD diagnosis? If it is after they have been informed they have CD, how much time has passed on average? What are the expectations of their following a GFD at the point of taking part in the study (I note in the ARCTIC paper it sounds as though it is an inclusion criteria for them to still be eating gluten? Is this true only at recruitment or also at data collection?)
Results: I don’t have a fundamental issue with so many analyses being conducted (i.e. in Table 2 and also the supplementary analyses), but I think some acknowledgement of possible multiple comparisons problems is needed. Could highlight in the text the results that would or would not survive e.g. bonferroni correction. Arguably this should also be acknowledged when raising these findings in the Discussion.
Discussion: CD patients who were not following the diet were excluded, do the implications of this need to be considered? It limits the validity of the type of CD patient that the results can be applied to.
Author Response
Comment 1 I have some specific suggestions below, but an overarching comment is that I am unclear about the patient’s medical advice. The paper is written as though we are to assume all (CD) volunteers will have been advised to follow the MD as part of their treatment. The language throughout is generally presumptive of this and classifying their diets as being “inadequate” etc. implies they are failing at following a guideline. I am not sure this is the case though? If the patients have all been told they should follow the MD this should be stated explicitly. If they have not then I think it reframes the paper’s objectives and would encourage the authors to adjust some of the introduction / discussion to make this less like patients may not be following advice, and more about this being a possible missed opportunity that the medical community should consider adopting properly in treatment approach.
Response 1
No dietary education related to the Mediterranean diet was provided in either group. When we talk about “adherence to the Mediterranean diet,” we are actually assessing the extent to which an individual’s diet aligns with the principles of the Mediterranean diet. This is the same approach present in all the publications cited on this topic.
An important message of our study is that although current guidelines for celiac patients emphasize that dietary consultations should focus on providing information about the gluten-free diet and ensuring adherence to it, compliance with the principles of the Mediterranean diet could offer additional health benefits to these patients.
Both current data and our own study indicate that even with strict gluten avoidance, the diet of celiac patients does not inherently meet the requirements for the prevention of cardiovascular and metabolic diseases. The ongoing RCT arm of our ARCTIC study is investigating the effects of a dietary intervention focused on the Mediterranean diet. The results of this study will provide high-level evidence on this topic, which could be incorporated into future guidelines.
Comment 2 Abstract: Additional methodological information would be desirable, particularly where the participants in the study came from and also the sample sizes.
Response 2
Thank you for the suggestion, we implemented changes in the abstract.
Comment 3 Methods (2.2): I appreciate the citation to more methodological information but please state in the current manuscript where the control group was recruited from. It would also be beneficial in the paper to state the headline exclusion criteria that apply to all participants (taken from the ARCTIC paper, e.g. heart failure, active cancers, refractory CD etc.).
Response 3
Thank you for your comment. We recruited healthcare workers and medical students who met the inclusion criteria for the control group. We also clarified this in the manuscript. We rephrased the following sentence: "Newly diagnosed CD patients (before starting a GFD) and CD patients on GFD for a minimum of 1 year were included in this study.”
Comment 4 I’d also like elaboration on the “newly diagnosed” group. At what point does study data collection occur with respect to their CD diagnosis? If it is after they have been informed they have CD, how much time has passed on average? What are the expectations of their following a GFD at the point of taking part in the study (I note in the ARCTIC paper it sounds as though it is an inclusion criteria for them to still be eating gluten? Is this true only at recruitment or also at data collection?)
Response 4
The patient enrollment process is detailed in the ARCTIC study protocol. Newly diagnosed patients were included after their diagnosis but before starting a gluten-free diet (GFD), and their adherence to the Mediterranean diet (MD) was assessed at this stage.
Following this, dietary counseling was provided, and only then did they begin the GFD. As a result, the MD adherence questionnaire for newly diagnosed, “naïve” patients fully reflects their dietary habits before the initiation of the GFD.
Comment 5 Results: I don’t have a fundamental issue with so many analyses being conducted (i.e. in Table 2 and also the supplementary analyses), but I think some acknowledgment of possible multiple comparison problems is needed. Could highlight in the text the results that would or would not survive e.g. Bonferroni correction. Arguably this should also be acknowledged when raising these findings in the Discussion.
Response 5
We did not make adjustments for multiplicity as these results serve for hypothesis-generating purposes. Alternatively, the analyses performed can be considered independent of each other. We acknowledged this in the limitation section of the manuscript.
Comment 6 Discussion: CD patients who were not following the diet were excluded, do the implications of this need to be considered? It limits the validity of the type of CD patient that the results can be applied to.
Response 6
Patients who do not adhere to the diet at all cannot be considered as receiving dietary treatment and were therefore excluded from the study. Anyone who provides care for a large number of celiac patients is likely to have a few who refuse to follow the diet entirely. Indeed, these patients are part of the full “real-world” spectrum. However, in a clinical study where dietary habits alongside a gluten-free diet are being assessed, such patients cannot be included. In our study, there was one such patient who was excluded.
Again, thank you for all the comments and suggestions made. Please, let us know if any further changes or improvements are required.